# Hybrids of Imatinib with Quinoline: Synthesis, Antimyeloproliferative Activity Evaluation, and Molecular Docking

**DOI:** 10.3390/ph15030309

**Published:** 2022-03-03

**Authors:** Carine Santos, Luiz Pimentel, Henayle Canzian, Andressa Oliveira, Floriano Junior, Rafael Dantas, Lucas Hoelz, Debora Marinho, Anna Cunha, Monica Bastos, Nubia Boechat

**Affiliations:** 1Laboratório de Sintese de Farmacos-LASFAR, Instituto de Tecnologia em Farmacos-Farmanguinhos, FIOCRUZ, Rua Sizenando Nabuco 100, Manguinhos, Rio de Janeiro 21041-250, Brazil; carineribeirost@gmail.com (C.S.); luiz.pimentel@far.fiocruz.br (L.P.); henaylecanzian@gmail.com (H.C.); andressa.oliveira@far.fiocruz.br (A.O.); lucas.hoelz@far.fiocruz.br (L.H.); debora.marinho@far.fiocruz.br (D.M.); monica.macedo@far.fiocruz.br (M.B.); 2Programa de Pós-graduação em Farmacologia e Química Medicinal do Instituto de Ciências Biomédicas–ICB-UFRJ, Centro de Ciências da Saúde-CCS, Bloco J, Ilha do Fundão, Rio de Janeiro 21941-902, Brazil; 3Laboratório de Bioquímica Experimental e Computacional de Fármacos, Instituto Oswaldo Cruz FIOCRUZ, Av. Brasil 4365, Manguinhos, Rio de Janeiro 21040-360, Brazil; floriano@ioc.fiocruz.br (F.J.); rafaeldantas11@hotmail.com (R.D.); 4Departamento de Química Orgânica, Campus do Valonguinho, Universidade Federal Fluminense–UFF, Niterói 24020-150, Brazil; annacunha@id.uff.br

**Keywords:** cancer, imatinib, PAPP, quinoline, 1,2,3-triazole, tyrosine kinase inhibitors

## Abstract

Imatinib (IMT) is the first-in-class BCR-ABL commercial tyrosine kinase inhibitor (TKI). However, the resistance and toxicity associated with the use of IMT highlight the importance of the search for new TKIs. In this context, heterocyclic systems, such as quinoline, which is present as a pharmacophore in the structure of the TKI inhibitor bosutinib (BST), have been widely applied. Thus, this work aimed to obtain new hybrids of imatinib containing quinoline moieties and evaluate them against K562 cells. The compounds were synthesized with a high purity degree. Among the produced molecules, the inhibitor 4-methyl-N3-(4-(pyridin-3-yl)pyrimidin-2-yl)-N1-(quinolin-4-yl)benzene-1,3-diamine (**2g**) showed a suitable reduction in cell viability, with a CC_50_ value of 0.9 µM (IMT, CC_50_ = 0.08 µM). Molecular docking results suggest that the interaction between the most active inhibitor **2g** and the BCR-ABL1 enzyme occurs at the bosutinib binding site through a competitive inhibition mechanism. Despite being less potent and selective than IMT, **2g** is a suitable prototype for use in the search for new drugs against chronic myeloid leukemia (CML), especially in patients with acquired resistance to IMT.

## 1. Introduction

Targeted therapy is the standard treatment for chronic myeloid leukemia (CML). It specifically interferes with target biomacromolecules related to carcinogenesis rather than acting in all rapidly dividing cells, unlike traditional chemotherapy [1]. In this context, BCR-ABL1 tyrosine kinase (TK), which is generated by chromosomal translocation between the ABL1 proto-oncogene of chromosome 9 and the BCR gene of chromosome 22 (Philadelphia chromosome), is a highly attractive molecular target for the development of selective chemotherapeutics for patients with CML [2]. The BCR-ABL1 protein is a constitutively active kinase that is not expressed in a healthy organism and allows cells to proliferate without regulation and become cancerous [3,4]. Thus, BCR-ABL1 TK inhibitors (TKIs) are the first-line therapy for most patients with CML, acting on the kinase domain by competing at the adenosine triphosphate (ATP)-binding site.

In the late 1980s, tyrphostin (**1**) was described as the first TKI (Figure 1A) [5], followed by imatinib (IMT), which presents phenylaminopyrimidine pyridine (PAPP) (Figure 1B) as the main pharmacophore group. Since then, the PAPP moiety has been used in the development of new TKIs to identify drugs more potent than IMT, such as nilotinib (NLT) (Figure 2) [6].

In a competitive way, IMT binds to the inactive form of BCR-ABL1, preventing protein binding to ATP. This can result in interruption of the substrate phosphorylation process and the transduction signal, inducing cell apoptosis [7,8]. Initial treatment preferably begins with IMT (first-line drug), mainly due to its safety profile [9]. Nonetheless, clinical resistance increases the need for developing new inhibitors, which are classified as second- and third-generation inhibitors. In cases of resistance, other second-generation drugs such as nilotinib (NLT) or dasatinib (DST) can be chosen. Although bosutinib (BST) is also a second-generation TKI, it is considered a second-line drug and is used only when IMT has failed [10]. Ponatinib (PNT) is the only representative third-generation inhibitor and is the frontier of TKI treatment for CML with the T315I mutation [11]. In Figure 2, the structures of these inhibitors are shown.

Currently, one of the greatest concerns related to the treatment of CML with TKIs is resistance related to mutations, which affect 20–30% of patients. Primary mutations occur at the beginning of treatment and are the most severe. Secondary mutations can develop over time. Another important issue is several toxic effects, such as hematological abnormalities, associated with the use of TKIs [12].

The resistance and toxicity associated with the use of TKIs highlight the importance of the search for new inhibitors. In this context, heterocyclic systems have been widely used in the rational design of new TKIs for CML [13].

Pyrimidine and its derivatives play important roles in many biological systems [14,15,16]. Different compounds containing the pyrimidine scaffold have been described, with an extremely high diversity of biological activities [17]. Among the TKIs used for CML, this nucleus is present in the IMT and NLT, composing the phenylamino-pyrimidine pyridine (PAPP) group as the main pharmacophore [18]. In addition, it is present in the DST scaffold, an important second-generation of TKI for CML.

The quinoline nucleus is present in several natural products, but it is frequently found in alkaloids. Some of these, such as cinchona, are still important as antimalarial and antiarrhythmic drugs [19]. Due to its versatility and diverse biological activity, medicinal chemists have explored it as a starting point for the rational design of new compounds that can act on new targets [19,20].

BST, a second-generation TKI, presents quinoline as one of the main pharmacophore fragments, which acts as a hydrogen bond acceptor group through the interaction of its nitrogen atom with methionine residue 318. Levinson and Boxer [21] performed molecular docking simulations between the enzymes BCR-ABL1 and BST, including other TKIs in which the quinoline nucleus was replaced by quinazoline. The results indicated that the presence of one additional nitrogen atom in quinazoline did not increase its selectivity for the BCR-ABL1 enzyme since no additional hydrogen bonding was observed. This finding showed the importance of quinoline and led to the development of several substances containing this nucleus, with the aim of producing new BST derivatives [6,22,23,24,25].

Therefore, we decided to explore the design and synthesis of new hybrids of imatinib with quinoline rings. Two series were proposed. In the first series, the PAPP fraction was directly linked to quinoline (**2a**–**h**). In the second series, different spacers were placed between the PAPP and quinoline fractions (**3a**–**b** and **4a**–**b**).

## 2. Design and Synthesis of Imatinib Hybrids 2a–h, 3a–b, and 4a–b

In the design of compounds **2a**–**h**, we investigated the introduction of groups with different electronic and stereo properties at positions 3, 5, 6, and 7 of the quinoline nucleus (Figure 3). Compounds **2a**–**d** maintained the nitrile group in the C-3 carbon of the quinoline ring [26], as in BST. Compounds **2e**–**f** were planned by replacing the nitrile group on carbon C-3 with an ethyl ester group. This substitution was used analogously to the work of Ghorab and coworkers, who synthesized quinoline derivatives containing the carboxamide group instead of the nitrile group [27].

Köprülü and coworkers prepared quinoline derivatives by varying both substituents and their positions. They found that some groups substituted at the C-2 carbon decreased the compound antiproliferative activity [28]. Based on these results, compounds **2g**–**h** were planned by omitting substituents on the C-3 carbon of the quinolinic ring. Compound **2h** is the only molecule that presents a PAPP skeleton connected to the C-2 carbon of quinoline, unlike the other compounds, in which this substitution occurred at the C-4 carbon. The interest in this position is associated with the fact that BST does not have a C-2 substituent. Furthermore, evidence indicates that quinolinic derivatives substituted at C-2 present antiproliferative activity [29,30]. Therefore, these changes are important for studying interactions other than those observed for the BST enzyme complex of the BCR-ABL1 enzyme and for evaluating the results of biological studies (Figure 3).

The substituents used on the C-6 carbon of quinoline were methoxy, present in bosutinib with the same substitution pattern of derivative **2c**, methyl, and chlorine. These last two were proposed with the objective of verifying the changes related to the electroding and withdrawing effects, respectively. Derivative **2d**, containing chlorine atoms at C-5 and C-7 carbons, was designed to verify the changes promoted by the removal of substituents from the C-6 carbon of the quinoline ring. The **2e**–**f** hybrids were designed with the exchange of the nitrile group on the C-3 carbon of the quinoline for an ethyl ester. As carbonyl is a bioisostere of nitrile, the ester was added to assess whether this exchange would influence the activity and toxicity of these hybrids (Figure 3).

In derivatives **2e**–**f**, the C-6 carbon of quinoline showed distinct substitution patterns, with the trifluoromethyl group (CF_3_) for compound **2f** and no substitution for **2e** (Figure 3). The choice of CF_3_ can be justified by the important pharmacokinetic characteristics of this group. Furthermore, the insertion of fluorine atoms in an organic compound can cause significant changes such as a decrease in metabolic lability; alteration of electronic characteristics; a possible increase in lipophilicity in relation to non-fluorinated molecules, which may favor passive diffusion through lipid barriers and, consequently, greater absorption in vivo [31].

In the design of compounds **3a**–**b**, the 1,2,3-triazole nucleus (shown in pink) was used as a spacer between the PAPP skeleton and the quinoline group. This spacer is described as a bioisostere of the amide functional group [32] (shown in purple) present at IMT. Various numbers of methylene carbons (shown in green) were included between the triazole and quinoline rings to analyze the contribution of this spacing to biological activity (Figure 3).

In the design of compounds **4a**–**b**, the amide functional group present in IMT was maintained, and the 1,2,3-triazole ring was used as an additional fragment to verify the effects of this insertion on antimyeloproliferative activity. The development of substances containing the 1,2,3-triazole nucleus with antitumor activity has been described in the literature [32,33]. With the same goal, various numbers of methylene carbons (shown in green) were included between the triazole and quinoline rings to analyze the contribution of this spacing to biological activity.

## 3. Results and Discussion

New hybrids **2a**–**h** were obtained by an aromatic nucleophilic substitution reaction between the PAPP group and substituted quinolines **5a**–**h** (Figure 1). Notably, quinolines **5g**–**h** were purchased commercially, and quinolines **5a**–**f** were synthesized. The synthetic route began with the preparation of phenylamine–acrylate derivatives **6a**–**f** by using a condensation reaction between anilines **7a**–**f** and acrylates **8a**–**f**. These compounds were obtained with yields between 86% and 96% [33]. Compounds **6a**–**f** were characterized by high-resolution mass spectrometry (HRMS), and the data were in agreement with the literature [34].

Quinolones **9a**–**f** were prepared by the thermal cyclization of derivatives **6a**–**f** using dowtherm, and the products were obtained in 65–80% yields. These compounds were characterized by HRMS, and the data were in agreement with the literature [34]. Derivatives **9a**–**f** were subjected to a chlorination reaction with phosphoryl chloride (POCl_3_), resulting in quinoline derivatives **5a**–**f** in 66–77% yields. Although these compounds are not new, they were characterized by ^1^H nuclear magnetic resonance (NMR) and HRMS techniques, and the data were in agreement with the literature [35].

The last step was an aromatic nucleophilic substitution between quinolinic intermediates **5a**–**h** and the PAPP group, which was acquired by donation from Cristália S.A. (Figure 1). Thus, the chemical structures of final products **2a**–**h** were identified by NMR, HRMS, and Fourier transform infrared (FTIR) techniques. ^1^H NMR spectra identified the hydrogens of the secondary amines, which presented as singlets with chemical shifts between 8.94 and 10.00 ppm. All analytical data, HRMS spectra, and high-performance liquid chromatography (HPLC) chromatograms of these compounds are presented in the Appendix A.

For the synthesis of hybrids **3a**–**b** (Figure 2), *N*-(5-azido-2-methylphenyl)-4-(pyridin-3-yl)pyrimidin-2-amine (**10**) was prepared by an aromatic nucleophilic substitution reaction through the formation of a diazonium salt using PAPP. This intermediate was obtained in a 95% yield [36]. Intermediates **11a**–**b** were prepared via 1,3-dipolar cycloaddition between compound **10** and the corresponding propargyl alcohol (**12a**) or pent-4-yn-1-ol (**12b**), which was catalyzed by copper I salt (Figure 2) [37]. These intermediates were obtained in suitable yields (84% and 76%, respectively). The characterization data for intermediate **11a** were in accordance with the literature [38]. For intermediate **11b**, the ^1^H NMR spectra showed a singlet at 8.32 ppm representing the H-5 hydrogen of the triazole ring. To obtain products **3a**–**b**, the last step was performed by aromatic nucleophilic substitution between intermediates **11a**–**b** and 4,7-dichoroquinoline in the presence of a base, with 73% and 44% yields, respectively. In the ^1^H NMR spectra, the absence of triplet signals at 5.32 ppm (*J* = 5.6 Hz) (**11a**) and 4.53 ppm (*J* = 5.7 Hz) (**11b**) attributed to the hydrogen of the hydroxy group demonstrated the formation of the ether bond with the quinolinic ring. The chemical structures of final products **3a**–**b** were identified by NMR, HRMS, and Fourier transform infrared (FTIR) techniques (see Appendix A).

The synthesis of final products **4a**–**b** (Figure 3) began with a substitution reaction between PAPP and chloroacetyl chloride to obtain the intermediate 2-chloro-*N*-(4-methyl-3-((4-(pyridin-3-yl)pyrimidin-2-yl)amino)phenyl)acetamide (**13**) (81% yield), and its characterization data were in accordance with the literature [39]. Subsequently, the conversion of **13** into 2-azido-*N*-(4-methyl-3-((4-(pyridin-3-yl)pyrimidin-2-yl)amino)phenyl)acetamide (**14**) occurred through a bimolecular nucleophilic substitution reaction with sodium azide in the presence of the sodium iodide catalyst. This compound was obtained in 85% yield, and its characterization data were in accordance with the literature [40]. The ^13^C NMR spectra of intermediate **14** showed methylene carbon signals at 50.6 ppm and carbonyl signals at 165.4 ppm. The next step involved click chemistry between azide **14** and the respective alcohols **12a**–**b**. Intermediates **15a**–**b** were obtained with 75% and 80% yields, respectively [37]. Through analysis of the ^1^H NMR spectra of both unpublished products **15a**–**b**, it was possible to identify singlets representing the hydrogen of the triazole ring of product **15a** at 7.99 ppm and that of intermediate **15b** at 7.88 ppm. Finally, these compounds were reacted via aromatic nucleophilic substitution with 4,7-dichloroquinoline to obtain **4a**–**b** in 32% and 20% yield (Figure 3) and were characterized by NMR. ^13^C NMR spectra showed the signals of the C-7′′ carbon of the quinolinic ring at 133.8 ppm for **4a** and 134.9 ppm for **4b**. The analytical data and HRMS spectra of all unpublished compounds, as well as HPLC chromatograms, are presented in the Appendix A. ^1^H NMR spectra showed the absence of both signals for hydroxyl groups and the presence of signals of the quinolinic ring. All analytical data, HRMS spectra, and HPLC chromatograms of the synthesized compounds are given in the Appendix A.

## 4. Biological Evaluation

Imatinib hybrids **2a**–**g**, **3a**–**b**, and **4a**–**b** were evaluated against K562 and WSS-1 cells, except derivative **2h**, which was in the purification phase and was evaluated only according to the concentration-response curve. K562 cells express the constitutively active BCR-ABL1 enzyme, and the WSS-1 cell line was used as healthy cells to analyze the selectivity of the derivatives under study.

In vitro testing by a cell viability assay was the first step in the evaluation of the antitumoral activity of the synthesized compounds using the K562 strain. For this purpose, resazurin (7-hydroxy-3*H*-phenoxazine-3-one-10-oxide), which is a faint fluorescent blue dye, was used as an indicator of redox reactions in these assays. When added to the culture, this substance enters the cytosol of the cells and is converted to the reduced form (resorufin). This process occurs via the activity of mitochondrial enzymes by capturing electrons from cofactors such as NADPH, FADH, and NADH, as well as cytochrome C. During the redox reaction, the dye changes from blue (oxidized form) to pink (reduced form), and this change can be measured by colorimetric reading [41].

The evaluation of series **2a**–**h** identified the **2g** derivative as the most promising compound, as it showed a higher percentage of cell viability inhibition than IMT and BST at 10 µM (Figure 4). It is worth mentioning that this is the only hybrid in the series without substitution at carbon C-3 of the quinoline ring. At this concentration, no significant differences in the percentage of inhibition were observed between the **2a**–**d** derivatives, which present the cyano group as a substituent at carbon C-3 of the quinoline, and compounds **2e**–**f**, with a diethylester substitution at carbon C-3 of this heterocycle. These results show that substitution at carbon C-3 of quinoline decreased activity in this series since the presence of substituents in this position resulted in the loss of activity, which was different from the results observed for BST.

Compounds **3a**–**b** and **4a**–**b** did not show promising activity in K562 cells, as at both tested concentrations, they were not able to significantly decrease cell viability (Figure 4). It is possible to observe that the presence of the amide group in molecules **4a**–**b** contributed to the anti-myeloproliferative activity upon comparison to the **3a**–**b** series. Both of the former molecules exhibited a higher percentage of inhibition of K562 cells at the tested concentrations (Figure 4). Additionally, the 1,2,3-triazole ring did not increase the antimyeloproliferative activity since the series of imatinib hybrids that contained this nucleus showed reduced activity.

Subsequently, a concentration-response curve was constructed for compounds **2g** and **2h**. The curves were processed with GraphPad Prism 8 software (GraphPad Software Inc., San Diego, CA, USA), and the CC_50_ (cytotoxic concentration to 50%) values of these compounds in K562 and WSS-1 cells were calculated. However, compound **2h** showed low solubility at concentrations above 30 μM, which could have directly interfered with the test results (Figure 5).

The results of the concentration-response curve in K562 cells for derivatives **2g**–**h** showed that the CC_50_ values were greater than that of IMT (CC_50_ = 0.08 µM) (Table 1). However, **2g** could be highlighted as the most promising (CC_50_ = 0.9 µM) and was seven-fold more potent than **2h** (CC_50_ = 6.2 µM). This result indicates that the substitution of PAPP on the C-2 carbon of quinoline reduced the activity of **2h**, as this was the largest structural difference among these derivatives. The evaluations of the concentration-response curve in WSS-1 cells for **2g**–**h** derivatives showed low CC_50_ values (0.2 and 4.9 µM, respectively) compared to IMT (CC_50_ = 8.9 µM) and, consequently, low selectivity indices (SI) (Table 1). These results indicated that these products were more toxic than the reference drug. In this work, the CC_50_ of BST could not be calculated by the applied method, but the literature describes a CC_50_ value lower than that of IMT in K562 cells. Nevertheless, it is not possible to compare values obtained in different evaluations [42].

## 5. Molecular Docking

We applied the molecular docking technique to predict the lowest-energy complexes between the most active synthesized inhibitors (**2g** and **2h**) and the BCR-ABL1 enzyme. Validation of the molecular docking protocol was carried out by the redocking of IMT and BST, which were cocrystallized with the BCR-ABL1 structure (PDB codes: 3PYY and 3UE4, respectively) [21,41]. Thus, the predicted poses with the lowest energy for IMT and BST presented MolDock scores of −206.02 and −332.36 arbitrary units (a.u.) and RMSD values of 1.68 Å and 1.85 Å, respectively, validating the docking protocol with RMSD values below 2.00 Å [43].

The molecular docking results showed that **2g** and **2h** had similar interactions with BCR-ABL1 to those described for IMT, presenting MolDock score values of −177.07 and −176.04 a.u., respectively (Table 2).

Additionally, the analysis of the interactions between IMT and BCR-ABL1 showed hydrogen bonds (H-Bond, blue dotted line) with Glu305, Thr334, Met337, and Asp400 (hydrogen bonding energy = −8.97 a.u.) and steric interactions with Asp400, and Ile379 (red dotted line) (steric binding energy = −193.65 a.u.) (Figure 6a), which were in agreement with the interactions found for IMT in the 3PYY crystal [44]. Analysis of the **2g**–**h**-BCR-ABL1 complexes showed interactions similar to those described for IMT (Figure 6a). Thus, compound **2g** presented hydrogen bonding interactions with Asp400, Met337, and Thr334 (hydrogen bonding energy of −7.34 a.u.) and steric interactions with Glu305, Leu389, Phe336, and Val318 (steric interaction energy = −169.73 a.u.) (Figure 6b). Similarly, compound **2h** showed hydrogen bonding interactions with Asp400 and Thr334 (hydrogen bonding energy = −5.31 a.u.) and steric interactions with Glu305, Ile312, Tyr272, and Val318 (steric interaction energy = −172.11 a.u.) (Figure 6c). Figure 7a,b presents the superpositions of the lowest energy of **2g** (best overlap) and **2h** with the cocrystallized IMT structure, respectively.

The molecular docking results showed that **2g** and **2h** had similar interactions with BCR-ABL1 to those described for BST (Table 3).

Analysis of the interactions between BST and the BCR-ABL1 enzyme showed a hydrogen bond only with the Met318 residue (hydrogen bond energy = −5.20 a.u.) and steric interactions with Met318, Thr315, Phe382, Asn322, Lys271, and Tyr320 (steric binding energy = −308.41 a.u.), which were similar to the interactions found for this drug in the 3UE4 crystal [21] (Figure 8a).

Analysis of predicted complexes between inhibitors **2g**–**h** and the BCR-ABL1 (3UE4) enzyme showed interactions different from those described for BST (Figure 8). Inhibitor **2g** showed hydrogen bonding interactions only with Thr315 (hydrogen bonding energy of −5.00 a.u.) and steric interactions with Thr315, Ala269, and Gly321 (steric interaction energy of −282.51 a.u.), with a MolDock score of −287.51 a.u. (Table 3; Figure 8b). However, inhibitor **2h** formed hydrogen bonds with Thr315 and Glu316 (binding energy hydrogen of −8.99 a.u.) and steric interactions with Thr315, Met318, Phe382, Ala308, Leu370, and Glu316 (steric interaction energy of −291.88 a.u.), with a MolDock score of −176.04 a.u. (Table 3; Figure 8c). Figure 9a,b presents the superpositions of the lowest energy of **2g** and **2h** with the cocrystallized BST structure, respectively.

Overall, the molecular docking results of the new PAPP-BST hybrids (**2g** and **2h**) suggested that the interaction with BCR-ABL1 occurs at the BST binding site through a competitive inhibition mechanism, presenting lower interaction energies than IMT. Additionally, among the synthesized compounds, the **2h** inhibitor displayed the lowest interaction energy when complexed to BCR-ABL1, suggesting a higher affinity for the enzyme than that of the **2g** inhibitor. However, these results cannot be directly correlated with in vitro antimyeloproliferative assessments, which require an enzyme inhibition assay against BCR-ABL1.

## 6. Conclusions

This work used the strategy of molecular hybridization between PAPP-based tyrosine kinase inhibitors involving the pharmacophoric group present in IMT and the heterocyclic system present in BST (quinoline nucleus). Twelve novel compounds (**2a**–**h**; **3a**–**b**; **4a**–**b**) were synthesized in suitable overall yields. All hybrids were evaluated for activity against the K562 and WSS-1 cell lines. Biological evaluations of final products **2a**–**h** in K562 cells identified **2g** as the most active of the series. In addition, it was observed that substitution at carbon C-3 of the quinoline group compromised the activity since the derivatives substituted at this position were inactive. In biological evaluations of the most active final products, **2g** and **2h**, in healthy cells (WSS-1), it was possible to observe low CC_50_ values (0.2 and 4.9 µM, respectively) compared to those of IMT (CC_50_ = 8.9 µM) and consequently lower selective indices.

Additionally, molecular docking studies propose that these two inhibitors (**2g** and **2h**) interact both at the binding site at the BST binding site and the IMT in BCR-ABL1. Regarding the BST binding site, compound **2h** presents a higher predicted affinity for the enzyme than compound **2g**. On the other hand, at the IMT binding site, compounds **2g** and **2h** showed an equivalent affinity for the enzyme. The hydrogen bond-type interactions present in the IMT-enzyme complex through its main pharmacophoric group (PAPP) were also observed in both compounds.

The proposed structural modifications were capable of generating compounds that inhibit the activity of K562 cells. These data suggest that it is possible to apply these compounds as prototypes for the development of potential drugs against CML, especially for use when patients develop acquired resistance to IMT. Therefore, further studies are being carried out to verify the biological activity of the compounds against cell lines resistant to IMT, along with enzymatic assays with the BCR-ABL1 enzyme.

## 7. Experimental Section

### 7.1. Synthesis

All solvents and reagents used in this work were of analytical grade. The reactions were monitored using thin-layer chromatography (TLC) with Merck TLC silica gel 60 F254 aluminum sheets 20 × 20 cm (Merck KGaA, Darmstadt, Germany). The eluents used were prepared volume by volume (*v*/*v*), and the TLC plates were visualized using a UV lamp (254 and 366 nm). For the reactions carried out using a microwave, the CEM Discovery reactor was used (CEM Corp., Matthews, NC, USA).

Melting points (m.p.) were determined using a Büchi Model B-545 apparatus and are uncorrected (Büchi Corporation, Flawil, Switzerland). Fourier transform infrared (FTIR) absorption spectra were recorded on a Thermo Scientific model Nicolet 6700 (Thermo Fisher Scientific, Waltham, MA, USA). Low-resolution mass spectra were obtained by electrospray ionization (MS-IES) in a Micromass ZQ4000 device (Waters, Milford, MA, USA). High-resolution mass spectrometry (HRMS) analyses were performed using a Bruker Compact (Q-TOF) (Brucker AG, Fällanden, Germany) device. ^1^H, ^13^C, and ^19^F nuclear magnetic resonance (NMR) analyses were performed with a Bruker HC device at 400 MHz for hydrogen, 100 MHz for carbon, and 376 MHz for fluorine (Bruker AG, Fällanden, Germany). Tetramethylsilane was used as an internal standard. The chemical shifts (δ) are reported in ppm, and the coupling constants (J) are reported in Hertz (Hz).

High-performance liquid chromatography (HPLC) analyses were conducted in a Shimadzu (VP) apparatus with pump models LC-20ADXR and LC-10AD, degassers DGU-20A5R and DGU-12A, detector by arrangement of photodiodes (DAFD) models SPD-M20A and SPD-M10A, automatic injectors SIL-30AD and SIL-10AD, and column ovens CTO-10A and CTO-20A. Data and control acquisitions were performed using Shimadzu CLASS-VP version 6.13 SP2 software. Separation was achieved with a Symmetry LC18 column measuring 50 × 2.1 mm, with a particle diameter of 3.5 μm and a maximum temperature of 80 °C. As the mobile phase, water was used as eluent (A), with a pH of 7.2 adjusted with ammonium hydroxide, and eluent (B) was methanol, with the elution gradient varying according to the method. The flow of the mobile phase ranged from 0.2 to 0.8 mL/min. For elementary analyses, a Perkin Elmer 2400 series II elementary analyzer was used.

### 7.2. General Procedure for the Preparation of Ethyl (Z)-3-(Phenylamino)acrylate (***6a**–**f***)

To a 50 mL monotubulated flask were added 30.86 mmol of the corresponding anilines **7a**–**f** (1 eq) and 30.86 mmol of the corresponding acrylates or malonates (**8a**–**f**) (1 eq) in 20 mL of ethanol. The mixture was kept under magnetic stirring and reflux for 2 h and 30 min. The completion of the reaction was evidenced by TLC using chloroform:methanol (9:1) as the eluent. The reaction mixture was poured into ice and, leading to the formation of a precipitate, which was subsequently filtered under vacuum to obtain product **6a** at 90% yield; m.p.: 130–134 °C; HRMS (ESI+) ([M+Na]+, *m/z*) theoretical value: 253.0947; found: 253.0948. **6b** = 92% yield; m.p.: 146–148 °C; HRMS (ESI+) ([M+ Na]+, *m/z*) theoretical value: 273.0401; found: 273.0392. **6c** = 96% yield; HRMS (ESI+) ([M+ Na]+, *m/z*) theoretical value: 269.0897; found: 269.0885. **6d** = 93% yield; HRMS (ESI+) ([M+ Na]+, *m/z*) theoretical value: 307.0011; found: 306.9996. **6e** = 96% yield; m.p.: 94–95 °C; HRMS (ESI+) ([M+Na]+, *m/z*) theoretical value: 286.1050; found: 286.1057. **6f** = 86% yield; m.p.: 95 °C; HRMS (ESI+) ([M+Na]+, *m/z*) theoretical value: 354.0924; found: 354.0938.

### 7.3. General Procedure for the Preparation of Quinolin-4(1H)-One (***9a**–**f***)

To a 100 mL monotubulated flask was added 40 mL of dowtherm, which was heated in a thermal blanket to reflux temperature (250 °C). Then, 30.86 mmol of the corresponding intermediates 6a–f were added separately, and the mixture was kept under magnetic stirring and reflux for 2 h. The product was poured into cold hexane, leading to the formation of a precipitate, which was subsequently vacuum filtered to obtain the desired products. **9a** = 77% yield; m.p.: >280 °C; HRMS (ESI+) ([M+Na]+, *m/z*) theoretical value: 207.0529; found: 207.0527. **9b** = 70% yield; HRMS (ESI+) ([M+Na]+, *m/z*) theoretical value: 226.9983; found: 226.9971. **9c** = 78% yield; HRMS (ESI+) ([M+Na]+, *m/z*) theoretical value: 223.0478; found: 223.0469. **9d** = 74% yield; HRMS (ESI+) ([M+Na]+, *m/z*) theoretical value: 236.9628; found: 236.9638. **9e** = 80% yield; m.p.: >250 °C; HRMS (ESI+) ([M+Na]+, *m/z*) theoretical value: 240.0631; found: 240.0633. **9f** = 65% yield; m.p.: 334–335 °C; HRMS (ESI+) ([M+Na]+, *m/z*) theoretical value: 308.0505; found: 308.0489.

### 7.4. General Procedure for the Preparation of 4-Chloroquinoline (***5a**–**f***)

To a 25 mL monotubulated flask were added 6.3 mmol of the corresponding quinolones **9a**–**f** and 15 mL of POCl_3_. The mixture was kept under magnetic stirring at reflux for 24 h. The completion of the reaction was evidenced by TLC using chloroform:methanol (9:1) as the eluent. The mixture was poured onto ice with constant stirring, neutralized with sodium hydroxide and extracted with dichloromethane (3 × 30 mL) to obtain the product. **5a** = 77% yield; HRMS (ESI+) ([M+H]+, *m/z*) theoretical value: 203.0370; found: 203.0372; ^1^H NMR (400 MHz; CD_3_COCD_3_, δ, ppm): 2.63 (s, 1H, H-6); 7.76 (dd, *J* = 8.7, 2.2 Hz, 1H, H-7); 8.06 (s, 1H, H-5); 8.08 (s, 1H, H-8); 8.90 (s, 1H, H-2). **5b** = 70% yield; HRMS (ESI+) ([M+H]+, *m/z*) theoretical value: 222.9824; found: 222.9831; ^1^H NMR (400 MHz; CD_3_COCD_3_, δ, ppm): 7.87 (dd, *J* = 9.0, 2.3 Hz, 1H, H-7); 8.14 (d, *J* = 9.0 Hz, 1H, H-8); 8.29 (d, *J* = 2.2 Hz, 1H, H-5); 8.96 (s, 1H, H-2). **5c** = 68% yield; HRMS (ESI+) ([M+H]+, *m/z*) theoretical value: 219.0320; found: 219.0311; ^1^H NMR (400 MHz; CD_3_COCD_3_, δ, ppm): 4.00 (s, 1H, H-6); 7.50 (d, *J* = 2.6 Hz, 1H, H-8); 7.70 (dd, *J* = 9.2, 2.8 Hz, 1H, H-7); 8.11 (d, *J* = 9.2 Hz, 1H, H-5); 9.04 (s, 1H, H-2). **5d** = 72% yield; HRMS (ESI+) ([M+H]+, *m/z*) theoretical value: 255.9362; found: 260.9592; ^1^H NMR (400 MHz; CD_3_COCD_3_, δ, ppm): 7.80 (d, *J* = 2.0 Hz, 1H, H-6); 8.13 (d, *J* = 2.0 Hz, 1H, H-8); 8.93 (s, 1H, H-2). **5e** = 70% yield; m.p.: 146–148 °C; HRMS (ESI+) ([M+H]+, *m/z*) theoretical value: 236.0473; found: 236.0482; ^1^H NMR (400 MHz; CD_3_COCD_3_, δ, ppm): 1.35 (t, *J* = 7.1 Hz, 3H, CH_3_); 4.31 (q, *J* = 7.1 Hz, 2H, CH_2_); 7.00 (d, *J* = 1.7 Hz, 1H, H-6); 7.07 (d, *J* = 1.7 Hz, 1H, H-8); 7.18 (m, 1H, H-7); 7.79 (d, *J* = 13.1 Hz, 1H, H-5); 8.33 (s, 1H, H-2). **5f** = 66% yield; HRMS (ESI+) ([M+H]+, *m/z*) theoretical value: 304.0347; found: 304.0326; ^1^H NMR (400 MHz; CD_3_COCD_3_, δ, ppm): 1.48 (t, *J* = 7.1 Hz, 3H, CH_3_); 4.53 (q, *J* = 7.2 Hz, 2H, CH_2_); 8.02 (dd, *J* = 8.8, 2.4 Hz, 1H, H-7); 8.28 (d, *J* = 8.8 Hz, 1H, H-8); 8.74 (s, 1H, H-5); 9.31 (s, 1H, H-2).

### 7.5. General Procedure for the Preparation of 4-Methyl-N^3^-(4-(pyridin-3-yl)pyrimidin-2-yl)-N^1^-(quinolin-4-yl)benzene-1,3-diamine (***2a**–**h***)

To a 5 mL monotubulated flask were added 0.39 mmol of the corresponding quinolines **5a**–**h**, 0.19 mmol of 3, and 0.78 mmol of potassium carbonate (1:1:2). The mixture was kept under magnetic stirring for 30 min in a microwave. The completion of the reaction was evidenced by TLC using chloroform:methanol (9:1) as the eluent. The mixture was washed with water to remove the salt and with toluene to remove some impurities. Purification was performed in a chromatographic column on silica gel using chloroform/methanol (99:1) as the eluent.

#### 7.5.1. 6-Methyl-4-((4-methyl-3-((4-(pyridin-3-yl)pyrimidin-2 yl)amino)phenyl)amino)quinoline-3-carbonitrile (**2a**)

Yield of 41%; m.p.: 237–238 °C; IR (cm^−1^): 1578, 1587, 2202, 3450; HRMS (ESI+) ([M+H]+, *m/z*) theoretical value: 444.1931; found: 444.1927; ^1^H NMR (400 MHz; CD_3_COCD_3_, δ, ppm): 2.31 (s, 3H, H-16); 2.51 (m, 3H, H-R6); 7.00 (dd, *J* = 8.4 e 2.0 Hz, 1H, H-15); 7.27 (d, *J* = 8.2 Hz, 1H, H-14); 7.39 (q, *J* = 4.8 Hz, 1H, H-26); 7.45 (d, *J* = 5.2 Hz, 1H, H-21); 7.69 (m, 2H, H-8, H-11); 7.83 (d, *J* = 8.5 Hz, 1H, H-7); 8.32 (s, 1H, H-29); 8.37 (dt, *J* = 8.1, 1.9 Hz, 1H, H-25); 8.52 (m, 2H, H-5, H-20); 8.66 (dd, *J* = 4.7, 1.4 Hz, 1H, H-27); 8.95 (s, 1H, H-2); 9.26 (d, *J* = 1.8 Hz, 1H, H-17); 9.71 (s, 1H, H-9). ^13^C NMR (100 MHz, CD_3_COCD_3_, δ, ppm): 17.76; 21.14; 87.72; 107.71; 116.89; 118.92; 119.58; 119.94; 121.84; 123.56; 128.52; 129.07; 130.44; 131.94; 133.66; 134.07; 136.03; 137.21; 138.27; 146.90; 148.06; 150.60; 151.32; 152.03; 159.35; 160.69; 161.33. HPLC (%, nm): 98.9 (344).

#### 7.5.2. 6-Chloro-4-((4-methyl-3-((4-(pyridin-3-yl)pyrimidin-2-yl)amino)phenyl)amino)quinoline-3-carbonitrile (**2b**)

Yield of 60%; m.p.: 251–254 °C; IR (cm^−1^): 1567, 1583, 2200, 3400, 787; HRMS (ESI+) ([M+H]+, *m/z*) theoretical value: 464.1385; found: 464.1371; ^1^H NMR (400 MHz; CD_3_COCD_3_, δ, ppm): 2.32 (s, 3H, H-16); 7.03 (dd, *J* = 8.0, 2.0 Hz, 1H, H-15); 7.29 (d, *J* = 8.1 Hz, 1H, H-14); 7.42 (q, *J* = 4.8 Hz, 1H, H-26); 7.46 (d, *J* = 5.2 Hz, 1H, H-21); 7.70 (d, *J* = 1.8 Hz, 1H, H-11); 7.86 (dd, *J* = 8.9, 2.0 Hz, 1H, H-8); 7.94 (d, *J* = 8.9 Hz, 1H, H-7); 8.39 (dt, *J* = 8.1, 1.8 Hz, 1H, H-25); 8.53 (d, *J* = 5.2 Hz, 1H, H-20); 8.67 (m, 2H, H-5, H-27); 8.27 (d, *J* = 1.8 Hz, 1H, H-29); 8.59 (s, 1H, H-2); 8.97 (s, 1H, H-9); 9.90 (s, 1H, H-17). ^13^C NMR (100 MHz, CD_3_COCD_3_, δ, ppm): 17.49; 87.93; 107.76; 116.42; 119.92; 120.03; 120.38; 122.16; 123.57; 129.09; 130.48; 130.95; 131.34; 131.94; 132.12; 134.12; 136.58; 138.33; 146.99; 148.04; 150.44; 151.31; 153.50; 159.34; 160.67; 161.34. HPLC (%, nm): 99.4 (344).

#### 7.5.3. 6-Methoxy-4-((4-methyl-3-((4-pyridin-3-yl)pyrimidin-2-yl)amino)phenyl)amino)quinoline-3-carbonitrile (**2c**)

Yield of 55%; m.p.: 256–261 °C; HRMS (ESI+) ([M+H]+, *m/z*) theoretical value: 460.1880; found: 460.1884; ^1^H NMR (400 MHz; CD_3_COCD_3_, δ, ppm): 2.31 (s, 3H, H-16); 3.91 (s, 3H, H-R6); 7.02 (dd, *J* = 8.0, 2.1 Hz, 1H, H-15); 7.28 (d, *J* = 8.1 Hz, 1H, H-14); 7.40 (q, *J* = 4.8 Hz, 1H, H-26); 7.47 (m, 2H, H-8, H-21); 7.68 (d, *J* = 2.0 Hz, 1H, H-11); 7.86 (m, 2H, H-7, H-20); 8.37 (dt, *J* = 8.0 e 1.8 Hz, 1H, H-25); 8.45 (s, 1H, H-2); 8.52 (d, *J* = 5.2 Hz, 1H, H-5); 8.66 (dd, *J* = 4.7, 1.4 Hz, 1H, H-27); 8.96 (s, 1H, H-9); 9.26 (d, *J* = 1.8 Hz, 1H, H-29); 9.64 (s, 1H, H-17). ^13^C NMR (100 MHz, CD_3_COCD_3_, δ, ppm): 17.76; 55.82; 87.92; 101.94; 107.72; 116.94; 119.71; 119.81; 120.08; 123.31; 123.57; 128.56; 130.47; 130.89; 131.93; 134.07; 137.32; 138.33; 144.14; 148.06; 150.12; 150.43; 151.33; 157.50; 159.35; 160.70; 161.34. HPLC (%, nm): 98,6 (344).

#### 7.5.4. 5,7-Dichloro-4-((4-methyl-3-((4-(pyridin-3-yl)pyrimidin-2-yl)amino)phenyl)amino)quinoline-3-carbonitrile (**2d**)

Yield of 55%; m.p.: 240–243 °C; IR (cm^−1^): 1550, 1576, 2200, 3444, 784; HRMS (ESI+) ([M+H]+, *m/z*) theoretical value: 498.0995; found: 498.0974; ^1^H NMR (400 MHz; CD_3_COCD_3_, δ, ppm): 2.28 (s, 3H, H-16); 6.91 (d, *J* = 7.5 Hz, 1H, H-15); 7.21 (d, *J* = 8.2 Hz, 1H, H-14); 7.43 (m, 2H, H-21, H-26); 7.61 (s, 1H, H-11); 7.85 (s, 1H, H-6); 8.00 (s, 1H, H-8); 8.38 (dt, *J* = 8.1, 1.8 Hz, 1H, H-25); 8.52 (d, *J* = 5.2 Hz, 1H, H-20); 8.66 (dd, *J* = 4.7, 1.5 Hz, 1H, H-27); 8.73 (s, 1H, H-9); 8.93 (s, 1H, H-2); 9.24 (d, *J* = 1.7 Hz, 1H, H-29); 9.44 (s, 1H, H-17). ^13^C NMR (100 MHz, CD_3_COCD_3_, δ, ppm): 17.76; 107.82; 116.14; 116.98; 117.03; 117.53; 123.70; 127.46; 127.46; 127.99; 129.25; 130.64; 130.96; 132.08; 134.22; 135.88; 138.25; 138.46; 148.18; 150.72; 151.44; 154.45; 159.55; 160.79; 161.37. HPLC (%, nm): 100 (344).

#### 7.5.5. Ethyl 4-((4-methyl-3-((4-(pyridin-3-yl)pyrimidin-2-yl)amino)phenyl)amino)quinoline-3-carboxylate (**2e**)

Yield of 26%; m.p.: 208–211 °C; IR (cm^−1^): 1545, 1571, 3426, 1780; HRMS (ESI+) ([M+H]+, *m/z*) theoretical value: 477.1993; found: 477.1981; ^1^H NMR (400 MHz; CD_3_COCD_3_, δ, ppm): 1.21 (t, *J* = 7.1 Hz, 3H, H-R3); 2.24 (s, 3H, H-16); 4.16 (q, *J* = 7.1 Hz, 2H, H-R3); 6.78 (dd, *J* = 8.1, 2.2 Hz, 1H, H-15); 7.17 (d, *J* = 8.3 Hz, 1H, H-14); 7.42 (m, 3H, H-6, H-21, H-26); 7.48 (d, *J* = 2.2 Hz, 1H, H-11); 7.74 (t, *J* = 8.2 Hz, 1H, H-7); 7.92 (dd, *J* = 8.4, 0.8 Hz, 1H, H-8); 8.02 (dd, *J* = 8.5, 0.7 Hz, 1H, H-5); 8.30 (dt, *J* = 8.5, 1.8 Hz, 1H, H-25); 8.47 (d, *J* = 5.2 Hz, 1H, H-20); 8.67 (dd, *J* = 4.8, 1.6 Hz, 1H, H-27); 8.92 (s, 1H, H-2); 8.98 (s, 1H, H-9); 9.22 (d, *J* = 1.7 Hz, 1H, H-29); 9.90 (s, 1H, H-17). ^13^C NMR (100 MHz, CD_3_COCD_3_, δ, ppm): 13.95; 17.63; 60.93; 107.75; 107.78; 116.72; 116.87; 119.92; 123.72; 125.15; 125.22; 127.27; 129.50; 130.79; 131.24; 132.09; 134.06; 138.62; 140.69; 148.09; 149.81; 150.01; 150.79; 151.43; 159.42; 160.83; 161.43. HPLC (%, nm): 99.2 (344).

#### 7.5.6. Ethyl 4-((4-methyl-3-((4-(pyridin-3-yl)pyrimidin-2-yl)amino)phenyl)amino)-6-(trifluoromethyl)quinoline-3-carboxylate (**2f**)

Yield of 32%; m.p.: 177–179 °C; IR (cm^−1^): 1572, 1572, 3445, 1785, 1307; HRMS (ESI+) ([M+H]+, *m/z*) theoretical value: 545.1907; found: 545.1893; ^1^H NMR (400 MHz; CD_3_COCD_3_, δ, ppm): 1.13 (t, *J* = 7.1 Hz, 3H, H-R3); 2.26 (s, 3H, H-16); 4.00 (q, *J* = 7.2 Hz, 2H, H-R3); 6.85 (dd, *J* = 8.1, 2.2 Hz, 1H, H-15); 7.21 (d, *J* = 8.2 Hz, 1H, H-14); 7.41 (q, *J* = 4.8 Hz, 1H, H-26); 7.44 (d, *J* = 5.2 Hz, 1H, H-21); 7.54 (d, *J* = 2.8 Hz, 1H, H-11); 8.00 (dd, *J* = 8.8, 1.7 Hz, 1H, H-8); 8.09 (d, *J* = 8.8 Hz, 1H, H-7); 8.32 (dt, *J* = 8.0, 1.8 Hz, 1H, H-25); 8.48 (d, *J* = 5.1 Hz, 1H, H-20); 8.59 (s, 1H, H-5); 8.66 (dd, *J* = 4.7, 1.5 Hz, 1H, H-29); 8.97 (m, 2H, H-2, H-9); 9.22 (d, *J* = 1.8 Hz, 1H, H-29); 10.05 (s, 1H, H-17). ^13^C NMR (100 MHz, CD_3_COCD_3_, δ, ppm): 13.73; 17.60; 60.88; 107.72; 108.43; 116.54; 116.76; 119.23; 122.56; 122.84 (q, *J* = 20.2 Hz, C-6); 123.55; 124.93; 125.26 (d, *J* =9.3 Hz, C-4); 126.32 (d, *J* = 11.5 Hz, C-8); 127.71; 130.84; 131.94; 133.95; 138.69; 139.78; 147.98; 149.35; 151.31; 153.14; 159.29; 160.67; 161.33; 166.62. HPLC (%, nm): 100 (344).

#### 7.5.7. 4-Methyl-N^3^-(4-(pyridin-3-yl)pyrimidin-2-yl)-N^1^-(quinolin-4-yl)benzene-1,3-diamine (**2g**)

Yield of 63%; m.p.: 153–157 °C; IR (cm^−1^): 1527, 1578, 3447, 768; HRMS (ESI+) ([M+H]+, *m/z*) theoretical value: 439.1432; found: 439.1429; ^1^H NMR (400 MHz; CD_3_COCD_3_, δ, ppm): 2.30 (s, 3H, H-16); 6.96 (d, *J* = 5.4 Hz, 1H, H-3); 7.07 (dd, *J* = 8.0, 2.2 Hz, 1H, H-15); 7.30 (d, *J* = 8.3 Hz, 1H, H-14); 7.44 (d, *J* = 5.2 Hz, 1H, H-21); 7.48 (q, *J* = 4.8 Hz, 1H, H-26); 7.56 (dd, *J* = 9.0, 2.2 Hz, 1H, H-5); 7.64 (d, *J* = 2.2 Hz, 1H, H-11); 7.87 (d, *J* = 2.2 Hz, 1H, H-8); 8.29 (d, *J* = 5.4 Hz, 1H, H-6); 8.40 (dt, *J* = 8.2, 1.8 Hz, 1H, H-25); 8.45 (d, *J* = 9.1 Hz, 1H, H-20); 8.54 (d, *J* = 5.1 Hz, 1H, H-2); 8.69 (dd, *J* = 4.7, 1.2 Hz, 1H, H-27); 9.02 (s, 1H, H-9); 9.13 (s, 1H, H-17); 9.26 (d, *J* = 1.8 Hz, 1H, H-29). ^13^C NMR (100 MHz, CD_3_COCD_3_, δ, ppm): 17.58; 101.27; 107.84; 117.95; 119.00; 119.18; 123.69; 124.37; 124.76; 127.29; 127.66; 130.92; 132.09; 133.84; 134.18; 137.50; 138.39; 148.03; 148.30; 149.14; 151.32; 151.51; 159.38; 160.96; 161.59. HPLC (%, nm): 98.1 (344).

#### 7.5.8. N^1^-(8-Chloroquinolin-2-yl)-4-methyl-N^3^-(4-(pyridin-3-yl)pyrimidin-2-yl)benzene-1,3-diamine (**2h**)

Yield of 81%; m.p.: 237–239 °C; IR (cm^−1^): 1527, 1592, 3440, 743; HRMS (ESI+) ([M+H]+, *m/z*) theoretical value: 439.1432; found: 439.1429; ^1^H NMR (400 MHz; CD_3_COCD_3_, δ, ppm): 2.24 (s, 3H, H-16); 7.14 (d, *J* = 8.9 Hz, 1H, H-14); 7.20 (m, 2H, H-3, H-6); 7.39 (m, 2H, H-5, H-7); 7.67 (m, 2H, H-11, H-26); 7.88 (dd, *J* = 8.3, 2.1 Hz, 1H, H-15); 8.08 (d, *J* = 9.0 Hz, 1H, H-21); 8.38 (dt, *J* = 8.0, 1.9 Hz, 1H, H-25); 8.51 (d, *J* = 5.2 Hz, 1H, H-20); 8.53 (d, *J* = 1.9 Hz, 1H, H-4); 8.61 (dd, *J* = 4.7, 1.6 Hz, 1H, H-27); 8.94 (s, 1H, H-9); 9.65 (s, 1H, H-17); 9.23 (d, *J* = 1.8 Hz, 1H, H-29). ^13^C NMR (100 MHz, CD_3_COCD_3_, δ, ppm): 17.42; 107.27; 113.45; 114.78; 114.85; 122.34; 123.50; 124.48; 125.02; 126.53; 129.08; 129.22; 129.96; 132.25; 134.22; 136.95; 137.83; 139.00; 143.00; 148.08; 151.07; 154.26; 159.29; 161.31; 161.53. HPLC (%, nm): 100 (344).

### 7.6. General Procedure for the Preparation of N-(5-Azido-2-methylphenyl)-4-(pyridin-3-yl)pyrimidin-2-amine (***10***)

To a 125 mL flask under refrigeration in an ice bath, 3.62 mmol (1003.6 mg) of PAPP (3) and 40 mL of distilled water were added. Using a funnel with a pressure equalizer, 3 mL of concentrated H_2_SO_4_ was slowly added into the mixture until complete solubilization of the PAPP in aqueous medium was achieved. Then, 5.42 mmol (379.1 mg, 1.5 eq.) of NaNO_2_ dissolved in 2 mL of water was added, and the reaction was kept under refrigeration in an ice bath. Afterward, 7.24 mmol (470.4 mg, 2 eq.) of NaN_3_ dissolved in 5 mL of distilled water was added under stirring. The mixture was protected from light and stirred for 3 h at room temperature. The end of the reaction was observed by TLC using a mixture of chloroform/methanol (9:1) as the eluent. Upon reaction completion, the mixture was neutralized to pH 7–8 with K_2_CO_3_. The mixture was extracted with dichloromethane (3 × 50 mL), and the organic phase was dried over anhydrous Na_2_SO_4_. The solvent was removed by evaporation, yielding azide **10** as a yellow solid, which, after drying, was stored in an amber flask (95% yield); m.p.: 114–116 °C; IR (cm^−1^): 2107, 1277; MS-ESI ([M+H]+, *m/z*): 302.25.

### 7.7. General Procedure for the Preparation of Alcohol (4-Methyl-3-((4-(pyridin-3-yl)pyrimidin-2-yl)amino)phenyl)-1H-1,2,3-triazol-4-yl) (***11a**–**b***)

To a 50 mL flask, 1 eq. of Compound 10 and 12 mL of an acetonitrile/water mixture (2:1) were added. The mixture was stirred, and 0.5 eq. of sodium ascorbate and 10% mmol CuSO_4_·5H_2_O were added. Then, 1.5 equivalents of appropriately substituted acetylene (**12a**–**b**) were added, and the mixture was irradiated in a microwave reactor. The reaction was monitored by TLC using a mixture of chloroform/methanol (95:5) as the eluent. The compounds were purified by recrystallization.

#### 7.7.1. 1-(4-Methyl-3-((4-(pyridin-3-yl)pyrimidin-2-yl)amino)phenil)-1H-1,2,3-triazol-4-yl)methanol (**11a**)

Condition: 8 min, 100W, 80 °C; recrystallized from acetonitrile; obtained as white solid; 84% yield; m.p.: 194–196 °C; IR (cm^−1^): 3358, 1010; HRMS (ESI+) ([M+Na]+, *m/z*) theoretical value: 382.1392; found: 382. 1384; ^1^H NMR (400 MHz; DMSO-d_6_, δ, ppm): 2.36 (3H, s, CH_3_); 4.62 (2H, d, *J* = 5.4 Hz, CH_2_); 5.32 (1H, t, *J* = 5.6 Hz, OH); 7.44 (1H, d, *J* = 8.3 Hz, H-5′); 7.51–7.59 (3H, m, H-6′, H-10′ e H-17′); 8.37 (1H, d, *J* = 2.1 Hz, H-2′); 8.51 (1H, dt, *J* = 8.0, 1.9 Hz, H-18′); 8.58 (1H, d, *J* = 5.2 Hz, H-11′); 8.64 (1H, s, H-5); 8.69–8.73 (1H, m, H-16′); 9.14 (1H, s, NH); 9.30 (1H, d, *J* = 1.5 Hz, H-14′). ^13^C NMR (100 MHz, DMSO-d_6_, δ, ppm): 17.20; 54.36; 108.16; 114.48; 114.85; 120.24; 123.26; 130.74; 130.95; 131.46; 133.79; 134.23; 138.41; 147.56; 148.42; 150.94; 159.01; 160.14; 161.02. CHN theoretical value %: (C) 63.50, (H) 4.77, (N) 27.28. Found %: (C) 63.49, (H) 4.74, (N) 27.91.

#### 7.7.2. 3-(1-(4-Methyl-3-((4-(pyridin-3-yl)pyrimidin-2-yl)amino)phenyl)-1H-1,2,3-triazol-4-yl)propan-1-ol (**11b**)

Condition: 20 min, 100W, 80 °C; recrystallized from acetonitrile; obtained as white solid; 76% yield; m.p.: 64.7–65.6 °C; IR (cm^−1^): 3401, 1039; HRMS (ESI+) ([M+Na]+, *m/z*) theoretical value: 410.1705; found: 410.1694; ^1^H NMR (400 MHz; DMSO-d_6_, δ, ppm): 1.77–1.87 (2H, m, CH_2_); 2.35 (3H, s, CH_3_); 2.68–2.80 (2H, m, CH_2_); 3.49 (2H, m, CH_2_); 4.54 (1H, t, *J* = 5.1 Hz, OH); 7.43 (1H, d, *J* = 8.4 Hz, H-5′); 7.51–7.57 (3H, m, H-6′, H-10′ e H-17′); 8.33 (1H, d, *J* = 2.1 Hz, H-2′); 8.48–8.52 (1H, m, H-18′); 8.53 (1H, s, H-5); 8.58 (1H, d, *J* = 5.2 Hz, H-11′); 8.71 (1H, dd, *J* = 4.7, 1.4 Hz, H-16′); 9.14 (1H, s, NH); 9.30 (1H, d, *J* = 1.8 Hz, H-14′). ^13^C NMR (100 MHz, DMSO-d_6_, δ, ppm): 17.18; 21.10; 31.50; 59.42; 107.65; 114.37; 114.77; 119.83; 123.24; 130.70; 130.84; 131.47; 133.78; 134.31; 138.38; 147.38; 147.56; 150.93; 159.01; 160.6; 161.03. CHN theoretical value %: (C) 65.10, (H) 5.46, (N) 25.31. Found %: (C) 64.98, (H) 5.47, (N) 25.29.

### 7.8. General Procedure for the Preparation of N-(5-(4-(((7-Chloroquinolin-4-yl)oxy))-1H-1,2,3-triazol-1-yl)-2-methylphenyl)-4-(pyridin-3-yl)pyrimidin-2-amine (***3a**–**b***)

To a 50 mL flask, 2 or 3 equivalents (224–336 mg) of potassium t-butoxide and 10 mL of DMF were added. The mixture was stirred under a N_2_ atmosphere, and 1 equivalent of alcohol (**11a**–**b**) dissolved in 5 mL of DMF was added. The reaction was kept at room temperature for 5 min, and a solution containing 1.5 equivalents of 4,7-dichloroquinoline dissolved in DMF was added. The mixture was held at room temperature under a N_2_ atmosphere for 3 or 24 h, and monitoring was performed by TLC using 95:5 chloroform/methanol as the eluent. Upon reaction completion, the mixture was poured into ice and neutralized with a 2 N aqueous hydrochloric acid solution.

#### 7.8.1. N-(5-(4-(((7-Chloroquinolin-4-yl)oxy)methyl)-1H-1,2,3-triazol-1-yl)-2-methylphenyl)-4-(pyridin-3-yl)pyrimidin-2-amine (**3a**)

Purification: recrystallized from acetonitrile; obtained as yellow solid; 73% yield; m.p.: 187–188 °C; IR (cm^−1^): 3419, 3271, 1119, 1073; HRMS (ESI+) ([M+H]+, *m/z*) theoretical value: 521.1605; found: 521.1604; ^1^H NMR (400 MHz; DMSO-d_6_, δ, ppm): 2.37 (3H, s, CH_3_); 5.57 (2H, s, CH_2_); 7.34 (2H, d, *J* = 5.3 Hz, H-3″); 7.48 (1H, d, *J* = 8.5 Hz, H-5′); 7.52 (2H, dd, *J* = 9.2, 5.0 Hz, H-10′ e H-17′); 7.56 (1H, dd, *J* = 8.9, 2.1 Hz, H-6″); 7.62 (1H, dd, *J* = 8.2, 2.3 Hz, H-6′); 8.01 (1H, d, *J* = 2.1 Hz, H-8″); 8.20 (1H, d, *J* = 8.9 Hz, H-5″); 8.40 (1H, d, *J* = 2.2 Hz, H-2″); 8.50 (1H, dt, *J* = 8.0, 1.9 Hz, H-18′); 8.58 (1H, d, *J* = 5.2 Hz, H-11′); 8.68 (1H, d, *J* = 3.9 Hz, H-16′); 8.82 (1H, d, *J* = 5.2 Hz, H-2″); 9.08 (1H, s, H-5); 9.17 (1H, s, NH); 9.30 (1H, s, H-14′). ^13^C NMR (100 MHz, DMSO-d_6_, δ, ppm): 18.30; 62.63; 103.14; 108.78; 115.82; 116.18; 119.79; 123.49; 124.50; 124.69; 126.81; 127.77; 131.88; 132.49; 134.84; 134.96; 135.05; 139.55; 143.36; 148.64; 149.72; 151.99; 153.56; 160.10; 160.73; 161.20; 162.09. HPLC (%, nm): 100 (227).

#### 7.8.2. N-(5-(4-(3-((7-Chloroquinolin-4-yl)oxy)propyl)-1H-1,2,3-triazol-1-yl)-2-methylphenyl)-4-(pyridin-3-yl)pyrimidin-2-amine (**3b**)

Purification: preparative chromatography-97: 3 chloroform/ methanol; obtained as yellow solid; 44% yield; m.p.: 107–108 °C; IR (cm^−1^): 3254, 2978, 1446, 1121, 1076; HRMS (ESI+) ([M+H]+, *m/z*) theoretical value: 549.1918; found: 549.1886; ^1^H NMR (400 MHz; DMSO-d_6_, δ, ppm): 2.33 (5H, m, OCH_2_CH_2_CH_2_- and CH_3_); 2.99 (2H, t, *J* = 6.1 Hz, OCH_2_CH_2_CH_2_); 4.37 (2H, t, *J* = 6.2 Hz, OCH_2_(CH_2_)_2_-); 7.06 (1H, d, *J* = 5.3 Hz, H-3″); 7.43 (1H, d, *J* = 8.4 Hz, H-5′); 7.49–7.57 (4H, m, H-6′, H-10′, H-17′ e H-6″); 7.97 (1H, d, *J* = 2.0 Hz, H-8″); 8.12 (1H, d, *J* = 8.9 Hz, H-5″); 8.20 (1H, d, *J* = 8.9 Hz, H-5″); 8.35 (1H, d, *J* = 1.7 Hz, H-2′); 8.49 (1H, dt, *J* = 8.0, 1.9 Hz, H-18′); 8.57 (1H, d, *J* = 5.2 Hz, H-11′); 8.63 (1H, s, H-5); 8.68 (1H, dd, *J* = 4.7, 1.5 Hz, H-16′); 8.74 (1H, d, *J* = 5.2 Hz, H-2″); 9.13 (1H, s, NH); 9.29 (1H, d, *J* = 1.7 Hz, H-14′). ^13^C NMR (100 MHz, DMSO-d_6_, δ, ppm): 18.25; 22.28; 28.40; 68.38; 102.59; 115.39; 115.77; 119.87; 124.23; 124.27; 126.66; 127.73; 131.76); 131.88; 132.51; 134.83; 135.33; 139.46; 147.73; 148.62; 148.63; 151.97; 153.57; 160.08; 161.20; 161.25; 162.07. HPLC (%, nm): 100 (254).

### 7.9. General Procedure for the Preparation of 2-Chloro-N-(4-methyl-3-((4-(pyridin-3-yl)pyrimidin-2-yl)amino)phenyl)acetamide (***13***)

To a 100 mL twin-tube flask under refrigeration in an ice bath and a nitrogen atmosphere were added 1.80 mmol (500 mg) of 3, 4.71 mmol (650 mg, 2.5 eq.) of anhydrous K_2_CO_3_ and 15 mL of anhydrous THF. A solution containing 0.3 mL (1.1 eq.) of chloroacetyl chloride dissolved in 15 mL of dry THF was added slowly through a pressure equalizer funnel. The reaction was held for 12 h at room temperature, and its completion was verified by TLC using a 95:5 mixture of chloroform/methanol as the eluent. After adding 20 mL of distilled water, the mixture was stirred for one hour. The solvent was concentrated under reduced pressure, and the mixture was filtered to obtain a yellow solid, which was washed with distilled water (81% yield); m.p.: 255.9–257.0 °C (degradation) (literature: 244.8 °C); IR (cm^−1^): 3350, 3288, 1662; MS-ESI ([M+H]+, *m/z*): 354.11.

### 7.10. General Procedure for the Preparation of 2-Azido-N-(4-methyl-3-((4-(pyridin-3-yl)pyrimidin-2-yl)amino)phenyl)acetamide (***14***)

To a 100 mL twin-tube flask under an inert atmosphere (N_2_), 0.50 mmol (176 mg) of 13, 0.75 mmol (48.5 mg, 1.5 eq.) of NaN_3_, 0.1 mmol (16.3 mg) of KI and 15 mL of anhydrous acetone were added. The mixture was held under reflux for 4 h, and monitoring was performed by TLC with chloroform as the eluent. After evaporating the solvent and washing the mixture with water, the mixture was filtered, yielding an orange solid, which was stored in an amber flask (85% yield); m.p.: 187–188 °C; IR (cm^−1^): 2103; HRMS (ESI+) ([M+Na]+, *m/z*) theoretical value: 383.1345; found: 383.1334; ^1^H NMR (400 MHz; DMSO-d_6_, δ, ppm): 2.21 (3H, s, CH_3_); 4.03 (1H, s, CH_2_); 7.18 (1H, d, *J* = 8.3 Hz, H-5); 7.29 (1H, dd, *J* = 8.2; 2.1 Hz, H-6); 7.44 (1H, d, *J* = 5.2 Hz, H-10); 7.50–7.57 (1H, m, H-17); 7.91 (1H, d, *J* = 1.9 Hz, H-2); 8.48 (1H, dt, *J* = 8.0, 1,9 Hz, H-18); 8.51 (1H, d, *J* = 5.1 Hz, H-11); 8.70 (1H, dd, *J* = 4.8; 1.5 Hz, H-16); 8.95 (1H, s, NH); 9.26 (1H, d, *J* = 1.8 Hz, H-14); 10.10 (1H, s, NH). ^13^C NMR (100 MHz, DMSO-d_6_, δ, ppm): 17.00; 50.66; 107.06; 114.91; 115.31; 123.22; 126.80; 129.70; 131.60; 133.85; 135.78; 137.37; 147.55; 150.79; 158.87; 160.45; 161.03; 165.45. CHN theoretical value %: (C) 59.99, (H) 4.48, (N) 31.09. Found: (C) 59.71, (H) 4.49, (N) 31.24.

### 7.11. General Procedure for the Preparation of 2-(4-(Hydroxy)-1H-1,2,3-triazol-1-yl)-N-(4-methyl-3-((4-(pyridin-3-yl)pyrimidin-2-yl)amino)phenyl)acetamide (***15a**–**b***)

To a 50 mL flask, 1 eq. of Compound 14 and 12 mL of an acetonitrile/water mixture (2:1) were added. The mixture was stirred, and 0.5 eq. of sodium ascorbate and 10% mmol CuSO_4_.5H_2_O were added. Then, 1.5 equivalents of appropriately substituted acetylene (**12a**–**b**) were added, and the mixture was irradiated in a microwave reactor. The reaction was monitored by TLC using a mixture of chloroform/methanol (95:5) as the eluent. The compounds were purified by recrystallization.

#### 7.11.1. 2-(4-(Hydroxymethyl)-1H-1,2,3-triazol-1-yl)-N-(4-methyl-3-((4-(pyridin-3-yl)pyrimidin-2-yl)amino)phenyl)acetamide (**15a**)

Condition: 60 min, 100W, 80 °C; recrystallized from acetonitrile; obtained as yellow solid; 75% yield; m.p.: 215–217 °C; IR (cm^−1^): 3372 and 3260, 1667, 1009; HRMS (ESI+) ([M+Na]+, *m/z*) theoretical value: 439.1607; found: 439.1588; ^1^H NMR (400 MHz; DMSO-d_6_, δ, ppm): 2.21 (3H, s, CH_3_); 4.54 (2H, d, *J* = 5.6 Hz, CH_2_OH); 5.22 (1H, t, *J* = 5.7 Hz, OH); 5.30 (2H, s, CH_2_); 7.19 (1H, d, *J* = 8.3 Hz, H-5′); 7.29 (1H, dd, *J* = 8.2; 2.0 Hz, H-6′); 7.44 (1H, d, *J* = 5.2 Hz, H-10′); 7.50 (1H, dd, *J* = 8.0, 4.8 Hz, H-17′); 7.94 (1H, d, *J* = 1.7 Hz, H-2′); 7.99 (1H, s, H-5); 8.46 (1H, dt, *J* = 8.0, 1.9 Hz, H-18′); 8.51 (1H, d, *J* = 5.1 Hz, H-11′); 8.69 (1H, d, *J* = 4.1 Hz, H-16′); 8.92 (1H, s, NH); 9.25 (1H, s, H-14′); 10.41 (1H, s, NH). ^13^C NMR (100 MHz, DMSO-d_6_, δ, ppm): 18.09; 52.61; 55.49; 108.17; 115.80; 116.14; 124.34; 124.84; 127.73; 130.84; 132.64; 134.97; 136.87; 138.49; 148.25; 148.60; 151.87; 159.94; 161.47; 162.10; 164.54. CHN theoretical value %: (C) 60.57, (H) 4.84, (N) 26.91. Found: (C) 60.45, (H) 4.84, (N) 26.88.

#### 7.11.2. 2-(4-(3-Hydroxypropyl)-1H-1,2,3-triazol-1-yl)-N-(4-methyl-3-((4-(pyridin-3-yl)pyrimidin-2-yl)amino)phenyl)acetamide (**15b**)

Condition: 80 min, 100W, 80 °C; recrystallized from acetonitrile; obtained as yellow solid; 80% yield; m.p.: 170–172 °C; IR (cm^−1^): 3270, 2939, 1670, 1009; HRMS (ESI+) ([M+Na]+, m/z) theoretical value: 467.1920; found: 467.1907; ^1^H NMR (400 MHz; DMSO-d_6_, δ, ppm): 2.22 (3H, s, CH_3_); 2.67 (2H, t, *J* = 7.6 Hz, CH_2_); 3.45 (2H, q, *J* = 6.2 Hz, CH_2_); 4.50 (1H, t, *J* = 5.1 Hz, OH); 5.27 (2H, s, CH_2_); 7.19 (1H, d, *J* = 8.3 Hz, H-5′); 7.28 (1H, dd, *J* = 8.2; 1.9 Hz, H-6′); 7.43 (1H, d, *J* = 4.9 Hz, H-10′); 7.88 (1H, s, H-5); 7.96 (1H, m, H-2′); 8.51 (2H, d, *J* = 7.9 Hz, H- 18′); 8.92 (1H, s, NH); 10.40 (1H, s, NH). ^13^C NMR (100 MHz, DMSO-d_6_, δ, ppm): 17.01; 21.04; 31.69; 51.51; 59.44; 107.17; 114.69; 115.02; 122.95; 126.59; 129.74; 133.58; 135.80; 137.41; 146.16; 147.67; 150.67; 158.86; 160.41; 161.21; 163.52. CHN theoretical value %: (C) 62.15, (H) 5.44, (N) 25.21. Found: (C) 62.08, (H) 5.45, (N) 25.36.

### 7.12. General Procedure for the Preparation of 2-(4-(((7-Chloroquinolin-4-yl)oxy))-1H-1,2,3-triazol-1-yl)-N-(4-methyl-3-((4-(pyridin-3-yl)pyrimidin-2-yl)amino)phenyl)acetamide (***4a**–**b***)

To a 50 mL flask, 2 or 3 equivalents (224–336 mg) of potassium t-butoxide and 10 mL of DMF were added. The mixture was stirred under a N_2_ atmosphere, and 1 equivalent of alcohol (**15a**–**b**) dissolved in 5 mL of DMF was added. The reaction was kept at room temperature for 5 min, and a solution containing 1.5 equivalents of 4.7-dichloroquinoline dissolved in DMF was added. The mixture was held at room temperature under a N_2_ atmosphere for 3 or 24 h, and monitoring was performed by TLC using 95:5 chloroform/methanol as the eluent. Upon reaction completion, the mixture was poured into ice and neutralized with a 2 N aqueous hydrochloric acid solution.

#### 7.12.1. 2-(4-(((7-Chloroquinolin-4-yl)oxy)methyl)-1H-1,2,3-triazol-1-yl)-N-(4-methyl-3-((4-(pyridin-3-yl)pyrimidin-2-yl)amino)phenyl)acetamide (**4a**)

Condition: 24 h; Preparative chromatography-94:6 chloroform/ methanol; obtained as yellow solid; 32% yield; m.p.: 245–247 °C; IR (cm^−1^): 3254, 1668, 1121, 1077; HRMS (ESI+) ([M+H]+, *m/z*) theoretical value: 578.1820; found: 578.1806; ^1^H NMR (400 MHz; DMSO-d_6_, δ, ppm): 2.22 (3H, s, CH_3_); 5.39 (2H, s, CH_2_); 5.51 (2H, s, OCH_2_); 7.19 (1H, d, *J* = 8.4 Hz, H-5′); 7.29 (1H, dd, *J* = 8.2, 2.1 Hz, H-6′); 7.33 (1H, d, *J* = 5.3 Hz, H-3″); 7.44 (1H, d, *J* = 5.2 Hz, H-10′); 7.47–7.51 (1H, m, H-17′); 7.56 (1H, dd, *J* = 8.9, 2.1 Hz, H-6″); 7.96 (1H, d, *J* = 1.9 Hz, H-2′); 8.00 (1H, d, *J* = 2.1 Hz, H-8″); 8.09 (1H, d, *J* = 8.9 Hz, H-5″); 8.43 (1H, s, H-5); 8.46 (1H, dt, *J* = 8.0, 1,9 Hz, H-18′); 8.51 (1H, d, *J* = 5.1 Hz, H-11′); 8.66 (1H, dd, *J* = 4.8, 1.6 Hz, H-16′); 8.80 (1H, d, *J* = 5.3 Hz, H-2″); 8.93 (1H, s, NH); 9.25 (1H, d, *J* = 1.7 Hz, H-14′); 10.47 (1H, s, NH). ^13^C NMR (100 MHz, DMSO-d_6_, δ, ppm): 17.01; 51.69; 61.45; 102.06; 107.10; 114.71; 115.06; 118.72; 123.20; 125.74; 126.17; 126.68; 126.71; 129.76; 131.56; 133.83; 133.88; 135.75; 137.43; 148.62; 150.75; 152.45; 158.86; 159.68; 160.39; 161.02; 163.31. HPLC (%, nm): 99.4 (254).

#### 7.12.2. 2-(4-(3-((7-Chloroquinolin-4-yl)oxy)propyl)-1H-1,2,3-triazol-1-yl)-N-(4-methyl-3-((4-(pyridin-3-yl)pyrimidin-2-yl)amino)phenyl)acetamide (**4b**)

Condition: 24 h; Preparative chromatography-95:5 chloroform/ methanol; obtained as yellow solid; 20% yield; m.p.: 145–146 °C; IR (cm^−1^): 3400, 3254, 2956, 1666, 1121, 1077; HRMS (ESI+) ([M+H]+, *m/z*) theoretical value: 606.2133; found: 606.2101; ^1^H NMR (400 MHz; DMSO-d_6_, δ, ppm): 2.22 (5H, m, OCH_2_CH_2_CH_2_- and CH_3_); 2.92 (2H, t, *J* = 7.4 Hz, OCH_2_CH_2_CH_2_); 4.32 (2H, t, *J* = 6.2 Hz, OCH_2_(CH_2_)_2_); 5.28 (2H, s, CH2); 7.03 (1H, d, *J* = 5.3 Hz, H-3″); 7.18 (1H, d, *J*= 8.4 Hz, H-5′); 7.27 (1H, dd, *J* = 8.2, 2.1 Hz, H-6′); 7.43 (1H, d, *J* = 5.2 Hz, H-10′); 7.48 (1H, m, H- 17′); 7.58 (1H, dd, *J* = 8.9, 2.1 Hz, H-6″); 7.98 (3H, s, H-2′, H-5 e H-8″); 8.15 (1H, d, *J* = 8.9 Hz, H-5″); 8.46 (1H, dt, *J* = 8.0, 1,9 Hz, H-18′); 8.51 (1H, d, *J* = 5.1 Hz, H-11′); 8.68 (1H, dd, *J* = 4.8, 1.6 Hz, H-16′); 8.73 (1H, d, *J* = 5.2 Hz, H-2″); 8.92 (1H, s, NH); 9.25 (1H, d, *J* = 1.7 Hz, H-14′); 10.41 (1H, s, NH). ^13^C NMR (100 MHz, DMSO-d_6_, δ, ppm): 18.09; 22.10; 28.59; 52.62; 68.30; 102.56; 108.17; 115.75; 116.08; 119.90; 124.35; 124.78; 126.72; 127.64; 127.73; 127.74; 130.82; 132.64; 134.85; 134.97; 136.86; 138.49; 146.30; 148.63; 149.65; 151.82; 153.58; 159.93; 161.27; 161.46; 162.08; 164.57. HPLC (%, nm): 96.3 (254).

## 8. Biological Evaluation

The assays were performed on two cell lines, as described below. K562 (ATCC^®^ CRL-243 ™): the cells used in this work were extracted from the bone marrow of a 53-year-old female patient who had CML (ATCC: The Global Bioresource Center b). WSS-1 (ATCC^®^ CRL2029 ™): the cells used in this work were a healthy human cell line obtained from the renal epithelium of a fetus and transformed with adenovirus 5 DNA (ATCC: The Global Bioresource Center c). The K562 strain was grown in RPMI-1640 medium (R8758, Sigma-Aldrich, St. Louis, MO, USA) according to the provider’s recommendations and the literature [27,28]. WSS-1 cells were cultured in high-glucose DMEM (Vitrocell) according to the provider’s recommendations (ATCC: The Global Bioresource Center c). The media were supplemented with 10% fetal bovine serum (FBS) and 50 μg/mL gentamicin. All cell lines were grown in a cell culture bottle with a 0.22 μm pore membrane filter on the lid, allowing the circulation of CO_2_. WSS-1 cells grew adherently, while cells of the K562 strain grew in suspension. All strains were maintained at 35 °C, at 5% CO_2_. In addition, all cell lines were routinely evaluated before freezing for storage in liquid nitrogen in the vapor phase for the detection of mycoplasma.

### 8.1. Cell Viability Analysis

For the evaluation of the cytotoxic actions of the synthesized compounds, resazurin (Sigma-Aldrich) was used at a concentration of 0.1 mg/mL. The resazurin salt (Alamar Blue^®^) has a blue, nonfluorescent color and can be reduced to resorufin, which is pink and highly fluorescent. This process is an indicator of cell proliferation and fluorescent/colorimetric cytotoxicity with redox properties.

### 8.2. WSS-1 Cell Assay

The first step of this assay consisted of plating the cells, with 5 × 10^4^ cells per well. In this experiment, black plates with transparent bottoms and 96 wells (Greiner Bio-One) were used, and after plating, the cells were returned to the incubator for a period of approximately 20 h to allow growth and cell adhesion. The next step consisted of adding the obtained compounds, followed by incubation for a period of 46 h. Then, resazurin was added at a final concentration per well of 0.01 mg/mL, and immediately, the first fluorescence reading (λex = 560 nm; λex = 590 nm) (zero time) was performed using a FlexStation 3 microplate reader (Molecular Devices). After the first reading, the plate was returned to the incubator, and after two hours, the second reading was performed, completing 48 h of incubation with the compounds. As a negative control, cells treated with DMSO at a concentration of 0.5% instead of the obtained compounds were used. Imatinib and bosutinib were used as positive controls. Commercial doxorubicin (Sigma-Aldrich) was also used as a positive control to determine the adequacy of the assay since it has recognized antineoplastic activity against several types of tumors. Preliminary screening of the compounds was carried out at fixed concentrations (1 and 10 μM) in triplicate. For the most active compounds, a concentration-response curve was constructed to obtain the CC_50_ values. It should be noted that each experiment was carried out at least in duplicate. The fluorescence value at time zero (time immediately after the addition of resazurin) in each well was subtracted from the values obtained after two hours of incubation with resazurin. The average value with zero-time subtraction was used as 100% cell viability and was applied to calculate the percentage of cell viability in each well.

### 8.3. K562 Cell Assay

Standardization of this assay was not carried out in this project; thus, the following ideal values were used for the execution of the experiment: quantity of cells of 2 × 10^4^ cell/mL, incubation time with the compounds of 48 h, incubation time with resazurin of one hour and DMSO used within the limit of 0.5%. As the cells grew in suspension, a long incubation period for adherence was not necessary. After plating, the cells were incubated for one hour, and then the prepared compounds were added and incubated for 47 h. After this period, resazurin was added at a final concentration per well of 0.01 mg/mL, and the first fluorescence reading (λex = 560 nm; λex = 590 nm) (zero time) was performed immediately using FlexStation 3 microplates (Molecular Devices). After the first reading, the plate was returned to the incubator, and after one hour, the second reading was performed, completing 48 h of incubation with the compounds. The compounds were solubilized in DMSO (Sigma-Aldrich) at concentrations ranging from 16.66 to 50 mM and maintained at −30 °C. The SI, which consists of the ratio between the CC_50_ values in normal cells and cancer cells (SI = CC_50_ normal cell/CC_50_ cancer cell), was used to quantify the safety margin of the newly synthesized compounds. For this calculation, it was necessary to construct the concentration-response curves of the selected compounds according to the screening.

### 8.4. Data Analysis and Graph Construction

Analyses of the data obtained and graph construction were performed with the aid of GraphPad Prism 6 software (GraphPad Software Inc., San Diego, CA, USA). The concentration-response curves and CC_50_ values were obtained using the four-parameter logistic model.

## Data Availability

Data is contained within the article and Appendix A.

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
