# Peer review of "Hybrids of Imatinib with Quinoline: Synthesis, Antimyeloproliferative Activity Evaluation, and Molecular Docking"

_pharmaceuticals, 2022, doi:10.3390/ph15030309_

Round 1

Reviewer 1 Report

This article reports th Hybrids of PAPP with quinoline its  synthesis, antimyeloproliferative activity and  docking studies authors should checked for the following

  1. Authors should mentioned the importance of quinoline and pyramidine derivatives in the introduction  author may get benefitted by the following references https://doi.org/10.1016/j.bmc.2020.115973 ,  https://doi.org/10.1016/j.bmcl.2019.126750 etc
  2. In scheme 2 and scheme 3 authors mentioned 76-84% and 70-75%. What side product is formed ?
  3. SAR need to be represented with figure and more no of compounds will help in better represntation of SAR.
  4. Mention the melting point of each solid compounds
  5. IR can be for each compounds.
  6. How is docking validated can be validated with 2-3 software
  7. NMR and Mass need to rechecked. some are not corelating and spectra can be given in the supporting documents

Reviewer 2 Report

this paper "Hybrids of PAPP with quinoline: synthesis, antimyeloproliferative activity evaluation, and molecular docking" was submitted to Pharmaceuticals for publication as research article.

this paper is interesting and chemistry very well described and conducted. The quality of paper for chemistry section is very high. nevertheless, some points must be highlighted before publication in Pharmaceuticals.

Pharmaceuticals is a high quality journal in medicinal chemistry/pharmacology sectors. this paper contains different compounds that are not very active in cancer cells. the SAR were not conducted in terms of optimization.

the antiproliferative activity must be performed in almost three leukemic cancer cells.

the interaction with the target must be experimentally validated bys using KO cells, antibodies, ecc.

the quality of figures must be improved.

Reviewer 3 Report

In this manuscript Santos et al, propose the use of a combination of PAPP and quinoline pharmacophores which are found in known BCR-ABL inhibitors, to provide early hits that could pave the way for the development of next generation inhibitors as drug candidates. Overall the manuscript is well written. The authors have systematically presented the design of molecules, their synthesis, evaluation of their cellular biological activity followed by molecular docking to support their observations. The characterization data reported for the compounds is extremely satisfactory.

Below are few suggestions to improve the quality of the manuscript –

  1. The idea behind the design and incorporation of linkers between the pharmacophore is clear and satisfactory. However, the number of analogs 2a-2g is limited and the choice of substitutions explored at R6 and R7 is not clear. The authors should provide a rationale of their choices. Further it is unclear why the quinoline with the chloride at C7 was used as a reagent for the synthesis of 3a-b and 4a-b, since both 2g and 2h do not have the particular quinoline core.
  2. A major concern is the poor selectivity of these compounds when evaluated in K562 Vs WSS-1 cells. The authors have evaluated the IC50 of the synthesized compounds in K562 cells for their cytotoxic activities, however these IC50 values do not show the particular target being engaged (BCR-ABL). It would be highly desired that the authors include IC50 values from a biochemical assay which will greatly strengthen the author’s claims that these compounds could be prototypes to new drugs in the future.
  3. Could the authors include 2h in figure 4? Or provide an explanation for why it was left out?

Below are few minor edits –

  1. Scheme II is missing azide reagent.
  2. On page 6, edit the first 2 sentences to properly reflect the chemistry being performed. There is no elimination reaction being performed.
  3. Figure 5 – representation of the IC50 values in two rows in each panel is confusing.
  4. Figure 5 legends is missing “2h”.
  5. On page 10 the authors list Figure 7a-c, there is no Figure 7c.
  6. Mass is usually reported as M+Na or M+H

Round 2

Reviewer 1 Report

comment incorporated

Author Response

We appreciate all suggestions and have amended our manuscript to improve the quality of the article to be more in line with the Pharmaceuticals.

Reviewer 2 Report

My suggestions were not used to improve the quality of the paper.

In this form, in my opinion the manuscript is not sufficiently suitable for publication, but if the editors agree, you could change the title and introduction in order to reduce the role of PAPP, which was not validated in the paper.

Author Response

please find the details in the attachment below.

Round 3

Reviewer 2 Report

the manuscript is now ready for publication